# An Insight into the Role of Reactant Structure Effect in Pd/C Catalysed Aldehyde Hydrogenation

**DOI:** 10.3390/nano12060908

**Published:** 2022-03-09

**Authors:** Marta Stucchi, Francesca Vasile, Stefano Cattaneo, Alberto Villa, Alessandro Chieregato, Bart D. Vandegehuchte, Laura Prati

**Affiliations:** 1Chemistry Department, University of Milan, Via Golgi 19, 20133 Milan, Italy; marta.stucchi@unimi.it (M.S.); francesca.vasile@unimi.it (F.V.); stefano.cattaneo2@unimi.it (S.C.); alberto.villa@unimi.it (A.V.); 2TotalEnergies Research Center—Qatar (TRCQ), Qatar Science & Technology Park, Al Gharrafa, Doha P.O. Box 9803, Qatar; alessandro.chieregato@totalenergies.com; 3TotalEnergies One Tech Belgium, Zone Industrielle Feluy C, B-7181 Seneffe, Belgium; bart.vandegehuchte@totalenergies.com

**Keywords:** ^13^C NMR relaxometry, T1 longitudinal relaxation time, Pd/carbon, aldehydes hydrogenation, benzaldehyde hydrogenation, octanal hydrogenation, carbon catalysts, solvent effect, heterogeneous catalysis, liquid-phase hydrogenation

## Abstract

The different activity of a 1% Pd/carbon catalyst towards aromatic and aliphatic aldehydes hydrogenation has been explored by ^13^C NMR relaxation. The ratio between T1 relaxation times of adsorbed (ads) and free diffusing (bulk) molecules (T1_ads_/T1_bulk_) can be used as an indicator of the relative strength of interaction between the reactant and the catalytic surface, where the lower the T1_ads_/T1_bulk_, the higher the adsorption strength. It can be seen that 1% Pd/carbon showed a reverse catalytic behaviour towards benzaldehyde and octanal hydrogenation, which can be explained by analysing the T1 relaxation times related to each substrate in the presence of the catalyst. Comparing and correlating the different T1_ads_/T1_bulk_ values, we were able to prove that the different catalytic results mainly depend on the contrasting adsorption behaviour of substrates on the catalyst. Moreover, the role of the solvent has been disclosed, as NMR results revealed that the adsorption of the reactants was strongly affected by the choice of solvent, which is revealed to be critical in modulating catalytic activity. As a consequence, T1_ads_/T1_bulk_ measurements can provide a guide to the selection of appropriate reaction conditions for improving catalytic activity.

## 1. Introduction

Predicting catalytic activities is of vital importance in catalyst design, considering the large efforts often required for synthesis and testing. The underlying phenomena responsible for catalytic performance is sometimes poorly understood due to limited catalyst characterisation. Microscopic and spectroscopic techniques are very often applied in the characterisation of heterogeneous catalysts, of which a significant number is based on supported metal nanoparticles. High-resolution transmission electron microscopy (HR-TEM), X-ray photoelectron spectroscopy (XPS), and Fourier-transform infrared spectroscopy (FT-IR), among other techniques, help untangling the role of these metal nanoparticles and the impact of their dimensions, surface exposure, oxidation state and interaction with the support on the overall catalytic performance. However, the whole catalytic behaviour is not only determined by the structural properties of the catalytic material, but is rather the result of the complex interplay between catalyst, reactant and reaction medium [1].

Some techniques allow studying the interaction between a molecule and the catalyst surface, as for example FT-IR spectroscopy of probe molecules [2] that are able to adsorb on specific sites which can subsequently be detected, identified and quantified, based on the intensity and position of the absorption band. Busca et al. [3], for example, studied the adsorption of formaldehyde on different oxides (TiO_2_, ZrO_2_, Al_2_O_3_) by FT-IR. This is of crucial interest for the understanding of many heterogeneous catalytic reactions, such as CO hydrogenation. However, FT-IR does not allow studying the competition between reactant and solvent in case of liquid-phase reactions. This is however possible by means of Attenuated Total Reflection spectroscopy (ATR-IR). For example, CO adsorption on Pt/Al_2_O_3_ and Pd/Al_2_O_3_ catalysts has been investigated with ATR-IR in gas and water, showing that water itself and pH have a large effect on the extent of CO interactions with the surface [4]. Moreover, operando ATR-IR analyses were performed on Pd/Al_2_O_3_ catalysts in contact with a solution of benzyl alcohol in cyclohexane, monitoring the restructuring of the catalyst surface and revealing important information about the metal active sites [5]. However, ATR-IR is limited to specific support and is not applicable to carbons.

A catalytic cycle is generally considered to be composed of different steps, where the adsorption/desorption of the reactant (R) (and the product (P)), as well as diffusion to the active site, are crucial in the observed activity of the catalyst. Figure 1 reports a simplified scheme:

Too weak adsorption usually leads to poor catalytic activity, while an extensively strong interaction induces an irreversible adsorption of (R) with a consequent deactivation of the active site. Moreover, the adsorption step is dramatically affected by the presence of a solvent and by other reaction conditions such as temperature and pressure. An important point to stress is that this step could be the common first step of different reactions.

Recently, it has been shown that the extent of interaction between the reactant and the catalyst can be evaluated through Nuclear Magnetic Resonance spectroscopy [6,7,8]. In particular, one of the NMR observables, the spin relaxation, is considered a sensitive probe for molecular dynamics. Relaxation times are divided into two types: longitudinal, which concerns change in magnetisation along the *z*-axis (T1), and transverse, which concerns a change in magnetisation in the *x*-*y* plane (T2). Relaxation properties, such as T1 and T2, may be used to identify molecular dynamics processes since they both depend on the rotational correlation time of the reactant [9]. Indeed, reduced T1 and T2 relaxation times are observed when liquid molecules adsorb on a solid surface due to a decrease in molecular mobility [10]. The ratio between T1 and T2 has been used in ^1^H-NMR spectroscopy, as a probe for the molecule/surface interaction [11,12], the energy of which is related to the residence time of molecules on the surface [13]. This methodology has been successfully used to study interactions between liquids and a variety of porous media [14] and has recently been extended to molecule-surface interactions in supported metal catalysts. D’Agostino et al. [8] studied the competition between substrate and solvent in the aerobic oxidation of 1,4-butanediol in CH_3_OH over Pt/SiO_2_, Pd/Al_2_O_3_ and Ru/SiO_2_ catalysts using the T1/T2 ratio. They showed that the catalysts with the lowest activity presented a stronger affinity for CH_3_OH than for 1,4-butanediol. It has also been shown that the addition of water even more inhibited the adsorption of the reactant over the catalyst surface, as the T1/T2 ratio of 1,4-butanediol decreased significantly when water was added [7].

However, as a result of the differences in magnetic susceptibility at the liquid–solid interface, and the limited chemical shift range associated with the ^1^H nucleus, the line broadening of the spectral resonance that occurs for adsorbed species can prevent the identification of individual ^1^H resonances. In the present paper, an alternative approach based on ^13^C NMR is presented showing some practical advantages. The larger chemical shift range of ^13^C allows to discriminate the individual resonances [15], thus the surface interaction of each individual carbon atom can be analysed. Additionally, concerning the coupling constant effect, proton decoupled ^13^C NMR does not suffer from spin coupling due to the very low natural abundance of ^13^C (1.1%). ^13^C NMR has been used to investigate oligomers grafted onto silica surfaces [16], ionic surfactants adsorbed on silica [17], and within catalysis, for example to study the adsorption of hydrocarbons on zeolites [18].

Moreover, as reported by Gladden et al. [15], in ^13^C NMR the longitudinal relaxation time T1 by its own can be a suitable probe for surface interactions, omitting the use of T2 which is more likely to suffer from resolution issues. The relatively slow molecular motion of the adsorbate on the adsorbent surface causes a decrease in T1 compared to its bulk phase. In this case, the ratio of T1 relaxation times for surface adsorbed (ads) to free diffusing (bulk) molecules (T1_ads_/T1_bulk_) can be used as an indicator of the relative strength of surface interaction. The lower the T1_ads_/T1_bulk_, the higher the adsorption strength.

In this study, we focus on Pd nanoparticles supported on a carbon support, commercially named GNP, which is a mesoporous graphitic carbon. This catalyst (1% Pd/GNP) has been used for the hydrogenation of aromatic (benzaldehyde) and aliphatic (n-octanal) aldehydes. Indeed, the catalytic hydrogenation/hydrogenolysis of aldehydes is of high interest from an industrial point of view and particularly relevant in the field of biomass transformation. Indeed, the high oxygen content present in the biomass must be removed to develop biomass-based processes and hydrogenation/hydrogenolysis catalytic processes represent one of the most appealing alternatives. Benzaldehyde and n-octanal can be considered model molecules that can be used to establish the influence of the aldehyde structure on the behaviour of the catalyst.

Alike, carbon-based catalysts are widely used in the industry because of their high stability in all the conditions required for the biomass transformation, and because they can combine hydrophobic (graphitic) and hydrophilic (oxygen functional groups) domains which could allow a better absorption of amphiphilic molecules.

While benzaldehyde could be readily converted at a temperature as low as 50 °C, negligible activity was observed with n-octanal. Here, we proved that by using ^13^C T1 NMR, this difference in catalytic behaviour can be predominantly attributed to the contrasting adsorption behaviour of both substrates on the catalytic surface which, in turn, is strongly affected by the choice of solvent. In fact, we disclosed that the presence of the solvent is critical in modulating the extent of adsorption, thus affecting catalysis, and our NMR results were able to guide the selection of more suitable reaction conditions.

## 2. Materials and Methods

### 2.1. Carbon Supports and Synthesis of Pd-Supported Catalyst

A commercial carbon xGnP^®^ (purchased by XG Sciences Inc., Lansing, MI, USA), with a BET surface area of 490 m^2^ g^−1^ and a total pore volume of 0.84 mL g^−1^ was used as support.

For the synthesis of Pd-supported catalysts, we used Na_2_PdCl_4_ (≥99.99%, Sigma-Aldrich, St. Louis, MO, USA) as the precursor, NaBH_4_ (powder, ≥98.0% Sigma Aldrich) as the reducing agent and Poly(vinyl alcohol) (PVA) (Mw 9000–10,000, 80% hydrolysed, Sigma Aldrich) as the protecting agent. All the reagents were used without further purification. The Pd/C catalyst was prepared by the sol-immobilization technique [19]. To an aqueous solution of Na_2_PdCl_4_ (0.5 mmol L^−1^) under constant stirring, we added a PVA solution (1% wt.) with a PVA-to-metal weight ratio of 0.5. A freshly prepared aqueous solution of NaBH_4_ (NaBH_4_-to-metal molar ratio of 8) was added to form a dark brown sol. After 30 min of sol generation, the colloid was immobilized by adding the carbon support (GNP) and acidified at pH 2 by sulfuric acid under continuous vigorous stirring. A metal loading of 1% wt. was targeted. After 1 h of stirring, the slurry was filtered, washed and dried at 80 °C for 2 h.

### 2.2. Catalytic Reaction and Conditions

The reference catalysts (1% Pd/GNP) were tested in the hydrogenation of both benzaldehyde (ReagentPlus^®^, ≥99% Sigma-Aldrich) and octanal (for synthesis, Sigma-Aldrich).

The reactions were carried out in a 100 mL stainless steel autoclave. We used 10 mL of a 0.3 M solution of substrate (benzaldehyde or octanal) in the solvent (*p*-xylene or dodecane, anhydrous, ≥99% Sigma-aldrich) and an amount of catalyst to reach a reactant-to-metal molar ratio of 1000:1. The reaction occurred at 50 °C and 2 bar of H_2_ pressure. Samplings were carried out by stopping the stirrer and quenching the reaction under cold water. In order to separate the catalyst, 200 µL of reaction mixture were withdrawn and centrifuged. Then, 100 µL of the supernatant solution was diluted with a solution of 1-dodecanol in *p-*xylene (external standard) for GC measurement.

Product analysis was carried out with a GC-MS (Thermo Scientific, Waltham, MA, USA, ISQ QD equipped with an Agilent VF-5 ms column, Santa Clara, California, USA) and the resulting fragmentation peaks were compared with standards present in the software database. Product quantification was carried out through a GC-FID equipped with a non-polar column (Thermo Scientific, TRACE 1300 equipped with an Agilent HP-5 column).

### 2.3. NMR Analyses

1,4-Dimethylbenzene-d10 (*p*-Xylene-d_10_), dodecane and chloroform-d (CDCl_3_) has been purchased from Sigma-Aldrich and used as solvents. The NMR tube has been prepared as follows. First, ~15 mg of the catalyst was loaded into the tube. Then, ~700 μL of the benzaldehyde or octanal solution (in *p*-Xylene-d_10_ or CDCl_3_) was added using a micropipette. The concentration of the aldehyde solution was set at a concentration as much as possible similar to that used in the catalytic tests.

NMR analyses were carried out on a Bruker Advance 600 MHz spectrometer at 298 K. For compound characterisation and ^13^C chemical shift identification, the J-modulated spin-echo sequence was used with 64k points in the time domain and 256 scans.

T1 measurements were obtained using standard inversion recovery from Bruker library (T1 measurements using inversion recovery with power gated decoupling).

One of the assumptions made in the T1 calculations is that the equilibrium is approached exponentially, and therefore, the magnetisation along the *z*-axis is represented by Equation (1).
M_z_ = M_0_(1 − e^−t/T1^)(1)

Herein, M_0_ is the magnetization at thermal equilibrium, t is the time elapsed, and T1 is the time constant that is obtained by plotting M_z_ as a function of time. The pulse sequence used in the inversion-recovery experiment is shown below:d1 − p1(180°) − d2 − p2(90°) − FID

In this experiment, the nuclei are first allowed to relax to equilibrium. A 180° pulse (p1) is then applied to invert the signals. The signals are then allowed to relax for a length of time (d2) that is varied in each experiment. After the variable d2 (recovery delay), a 90° pulse (p2) was applied, and the FID was recorded. The FID records the spectrum intensity as a function of the variable delay d2. The signal will have relaxed more with longer d2. The peak intensity will reflect the extent to which each signal has relaxed during the d2 period.

14 T1 recovery delays were used ranging from 0.05 ms to 100 s. We used 8 repeat scans and a relaxation delay (d1) of 50 s between each scan to ensure a maximum signal (and to ensure the reach of the equilibrium, the d1 delay in the pulse sequence should be set to ~5 × the longest T1 of interest in the molecule) was maintained at all times.

The analysis of the T1 measurements was performed with the standard Bruker routine for T1/T2 calculation and with the Bruker Dynamic Centre software version 2.5.6, using the following fitted function:f(t) = Io × [1 – a × exp (−t/T1)]
where I_0_ is the equilibrium magnetization and the parameter a determine the magnetization at time zero, that thus corresponds to I_0_(1 − a).

The errors associated with fitting the bulk and adsorbed T1 were all within ±1%.

## 3. Results and Discussion

### 3.1. Catalytic Results

As the reference catalysts, 1 wt% Pd supported on GNP (1% Pd/GNP) was. GNP is a commercial carbon support consisting of graphitic nanoplates with a specific BET surface area, pore volume and oxygen surface functionalization (Table 1).

As already mentioned, for the catalyst preparation, Pd nanoparticles were synthesized and loaded onto the carbon surface by the sol-immobilisation method [19], obtaining the final catalyst 1% Pd/GNP. The catalyst was characterized by TEM showing metal particle dimension and distribution of 3.5 ± 1.1 nm (Figure 2) and then by XPS (Table 2) showing a Pd^0^ and Pd^2+^ relative abundance of 68% and 32%, respectively. No variation of Surface Area nor pore volume have been observed after Pd particle deposition.

The catalytic activity was evaluated in benzaldehyde and octanal hydrogenation for comparison between an aromatic and an aliphatic aldehyde using the same catalyst, thus allowing considering only the effect of the different substrate.

#### Benzaldehyde and Octanal Hydrogenation in *p-*Xylene

The hydrogenation reactions were performed under mild conditions (50 °C, 2 bar of H_2_ and a substrate-to-metal ratio of 1000:1), using first *p-*xylene as solvent. At these conditions, aromatic solvents are usually used because of their high boiling point; here, *p-*xylene was chosen in particular because it does not convolute the analysis of by-products such as toluene. The reaction pathways are shown in Figure 3: it is generally accepted that first the carbonyl group is hydrogenated to alcohol, followed by hydrogenolysis of the C-O bond forming the hydrocarbon.

The bare carbon supports were not active in the catalytic hydrogenation of both aldehydes. On the contrary, 1% Pd/GNP almost fully converted benzaldehyde within 1 h (95% conversion), with the main reaction product being benzyl alcohol. Benzyl alcohol was then converted into toluene after 2 additional hours. Figure 4 shows the reaction profile.

Under the same reaction conditions, 1% Pd/GNP was not active towards octanal hydrogenation. In fact, no octanal conversion was observed, even after an extended reaction time (5 h). The reaction temperature and hydrogen pressure were then increased up to 150 °C and 20 bar, respectively, but no products of hydrogenation or hydrogenolysis were observed also under these conditions (Table 3).

### 3.2. T1_ads_ T1_bulk_ ^13^C NMR Analyses

NMR analyses and corresponding T_ads_/T1_bulk_ calculations were performed on samples composed of either benzaldehyde or octanal, on 1% Pd/GNP catalysts. Additionally, the solvent itself has been evaluated by NMR, in order to disclose its role in modulating the adsorption of above substrates.

The T1_ads_/T1_bulk_ ratio has been used as indicator of the relative strength of surface interaction, where the lower the T1_ads_/T1_bulk_ ratio the lower the mobility and hence, strong adsorption. In particular, a T1_ads_/T1_bulk_ ratio equal or very close to 1 indicates that there is no interaction and thus no adsorption of the substrate on the catalysts, as well as T1_ads_/T1_bulk_ values ≥ 0.9 indicate a very weak interaction and negligible adsorption.

#### 3.2.1. Benzaldehyde Adsorption

The adsorption of benzaldehyde was studied on the bare carbon support (GNP) and on the corresponding Pd-supported catalyst (1% Pd/GNP).

We calculated the T1 relaxation time in the presence of the catalyst (T1_ads_) and of the free diffusing benzaldehyde (T1_bulk_). We were able to discriminate four different signals related to specific carbon atoms within the benzaldehyde structure, according to Figure 5.

As a lower T1_ads_/T1_bulk_ ratio indicates a stronger benzaldehyde adsorption [15], the mean adsorption energy is mostly higher on the Pd-supported catalyst than on bare carbon (Table 4), due to the stronger adsorption of benzaldehyde, possibly due to the presence of Pd NPs acting as binding sites. Moreover, looking at the T1 values of the individual carbon atoms on Pd/GNP, we were able to identify the specific adsorption mode of benzaldehyde. The aldehydic carbon (C^1^, Table 4) showed a stronger adsorption on the Pd/GNP surface (T1_ads_/T1_bulk_ = 0.65) compared to the others; thus, we hypothesized that benzaldehyde approaches the surface of Pd/GNP with an orthogonal configuration, and not with a flat configuration (Figure 6) as depicted by DFT calculation [20] on different type of carbon support [20].

#### 3.2.2. n-Octanal Adsorption

As in the case of benzaldehyde, by NMR we investigated the adsorption of octanal on the Pd-supported catalyst. We calculated the T1 relaxation time for surface adsorbed octanal (T1_ads_) and the T1 of the free diffusing octanal (T1_bulk_) (Table 5). We were able to identify 7 different signals related to specific single carbon atoms within octanal, according to Figure 7.

The mean T1_ads/_T1_bulk_ ratio indicates a very weak adsorption of octanal on both the bare GNP and the Pd/GNP (Table 5), with the mean T1_ads/_T1_bulk_ values exceeding 0.90.

The overall conclusion based on these data is that the poor catalytic activity exhibited by Pd/C catalysts in the case of n-octanal is possibly due to a weak interaction of n-octanal on the catalyst surface. However, at that point the underlying reason of this remarkable difference with benzaldehyde was not clear.

### 3.3. Solvent Effect

A possible explanation for the earlier results is related to a competitive adsorption of the solvent on the active sites. Therefore, we investigated the impact of the solvent, i.e., *p-*xylene (see the paragraph 3.2 related to the catalytic results) on the adsorption of both substrates. We calculated the T1 relaxation time for *p-*xylene in the presence of the catalyst (T1_ads_) and the T1 of the free diffusing *p-*xylene (T1_bulk_). We were able to discriminate all signals related to the three different carbon atom environments within *p-*xylene, according to Figure 8. Results are reported in Table 5.

The mean T1_ads/_T1_bulk_ ratio (0.82–0.83) confirmed that *p-*xylene is adsorbed on bot bare GNP and Pd/GNP. The lowest values of T1_ads_/T1_bulk_ were obtained for C^2^ (Table 6, second line), while the highest values of T1_ads_/T1_bulk_ are those related to the carbon atom of the methyl group (C^1^), indicating its weaker adsorption compared to the aromatic carbon atoms.

With the aim of evaluating the competition in adsorption between *p-*xylene (the solvent) and the reactants (benzaldehyde and octanal), we compared the T1_ads_/T1_bulk_: on Pd/GNP, benzaldehyde showed a T1_ads_/T1_bulk_ mean value of 0.82 (Table 4), close to 0.83 obtained for *p-*xylene (Table 6). In contrast, considering the T1_ads_/T1_bulk_ for *p-*xylene and octanal in *p-*xylene, the T1_ads_/T1_bulk_ ratio of octanal was 0.93 on Pd/GNP (Table 5), which was always higher than the corresponding value for *p-*xylene, 0.83 (Table 6). Therefore, the lower T1_ads_/T1_bulk_ ratio of *p-*xylene showed that it is more strongly adsorbed on the catalyst surface, disfavouring the adsorption of octanal.

We supposed that the strong interaction of *p-*xylene with the catalyst was due to its aromatic nature. Therefore, we tested a non-aromatic solvent such as dodecane measuring the corresponding T1_ads/_T1_bulk_ (Table 7). In this case, we are able to measure only the carbon of the CHO group being the other C atoms signals superimposed to those of dodecane. The value clearly confirmed a higher adsorption of n-octanal (0.85 versus 0.98). Interestingly, despite the presence of a non-aromatic solvent, the T1_ads/_T1_bulk_ of octanal remains 1 on bare GNP showing that the presence of Pd nanoparticles is crucial to have the adsorption of octanal, as obtained in the case of benzaldehyde (see Table 4).

However, as stated above, the presence of dodecane does not allow measuring the signals of each carbon atom within the carbon chain of octanal. Therefore, we also performed some analyses using CDCl_3_ (Figure 9), the most common solvent for NMR analysis (Table 8), hypothesizing that its adsorption could be negligible as observed with dodecane, as its affinity to carbon is reported not very high [21].

As expected, a T1_ads_/T1_bulk_ ratio superior to 0.9 indicates a weak adsorption of CDCl_3_ on both bare GNP and Pd/GNP (Table 8a). We then measured the T1_ads_ and the T1_bulk_ of octanal (Table 8b) dissolved in CDCl_3_, discriminating all individual C atom signals.

Looking at the T1_ads_/T1_bulk_ values obtained with bare GNP in CDCl_3_, we observed that, as each single T1_ads_/T1_bulk_ value is always higher than 0.90 (Table 8b), octanal is not adsorbed on the bare carbon. However, when Pd is present, the T1_ads_/T1_bulk_ values are always below 0.87 (Table 8b), thus, as in the case of dodecane, Pd sites are crucial for adsorption.

Therefore, and accordingly with the NMR results, we verified the importance of the adsorption step in catalytic activity by carrying out octanal hydrogenation experiments on Pd/GNP in the presence of dodecane and comparing the results with those earlier obtained in *p-*xylene.

As reported in Table 9, by changing the solvent from *p-*xylene to dodecane, we were able to reach a significant conversion of octanal in the presence of 1% Pd/GNP. Strikingly, the selectivity of the reaction was different than by using dodecane. Octanal was converted mainly to dioctyl ether (Table 9), probably formed via consecutive reduction and hydrogenolysis reactions. Therefore, considering that a change in the solvent could also affect other aspects related to the catalytic reaction, we investigated the behaviour of Pd/GNP as catalyst for benzaldehyde hydrogenation using dodecane (Figure 10b) as the solvent instead of *p-*xylene (Figure 10a).

In the case of benzaldehyde, the change in the solvent resulted in a more moderate change in reaction profile (Figure 10b). In particular, when dodecane was used as the solvent, toluene is apparently formed from the beginning of the reaction, whereas in the case of *p-*xylene it seemed to be formed subsequently to benzyl alcohol formation. In this case, a possible explanation is that the absence of competitive adsorption with the solvent enhanced the residence time of adsorbed benzaldehyde and also benzyl alcohol on the surface, thus favouring direct toluene production.

## 4. Conclusions

The different activity of 1% Pd/GNP catalyst towards aromatic and aliphatic aldehydes hydrogenation has been explored by ^13^C NMR relaxation.

By studying the relaxation time ratio T1_ads_/T1_bulk_, we can correlate the different catalytic behaviour of benzaldehyde and n-octanal with their adsorption strength on the catalyst.

The choice of the solvent is crucial in the adsorption of the reactant, which is a pre-requisite for catalytic activity. When *p-*xylene was used as the solvent, benzaldehyde can be adsorbed on the catalyst surface and subsequently converted, whereas octanal resulted inert due to *p-*xylene governing the adsorption step. Further NMR analysis demonstrated that octanal can adsorb and react on Pd on GNP catalyst when a non-aromatic solvent is used. Therefore, as final proof of concept, a catalytic test changing the solvent from *p-*xylene to dodecane has been performed showing that 1% Pd/GNP was able to hydrogenate 40% of octanal in 2 h. It was also disclosed that the selectivity of the reaction can be affected also by the solvent: when dodecane was used as the solvent, n-octanal was converted to dioctyl-ether, whereas benzaldehyde showed a faster production of toluene.

At present, T1_ads_/T1_bulk_ measurements do not allow to differentiate physical vs. chemical adsorption but were able to clearly indicate the role of the Pd active site in the adsorption of the substrate on the catalyst surface, which in turn should be considered a pre-requisite for having catalytic activity. Further studies are ongoing in order to fine-tune the technique in an attempt to disentangle chemical from physical adsorption.

## Figures and Tables

**Figure 1 nanomaterials-12-00908-f001:**
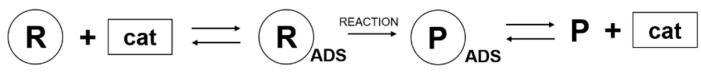
The catalytic cycle, ([ADS] = physically or chemically adsorbed).

**Figure 2 nanomaterials-12-00908-f002:**
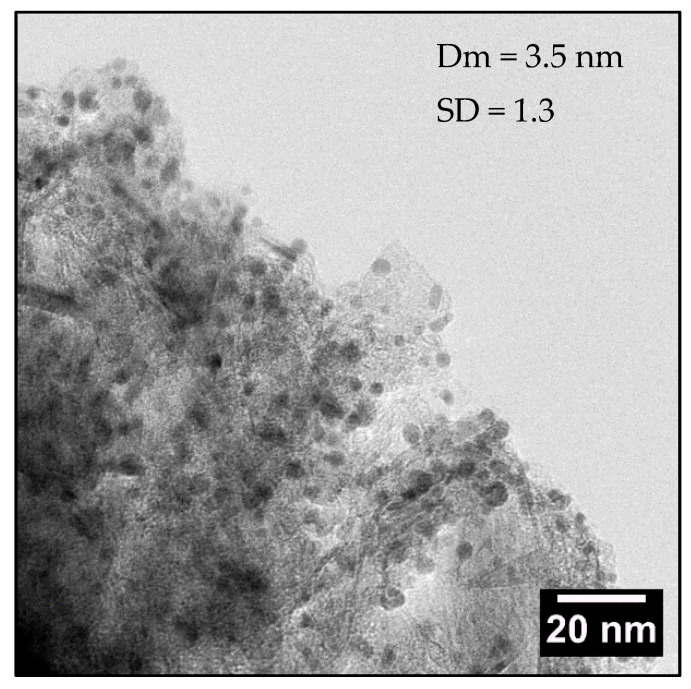
TEM analysis with relative mean diameter (Dm) and standard deviation (SD) of 1% Pd/GNP.

**Figure 3 nanomaterials-12-00908-f003:**
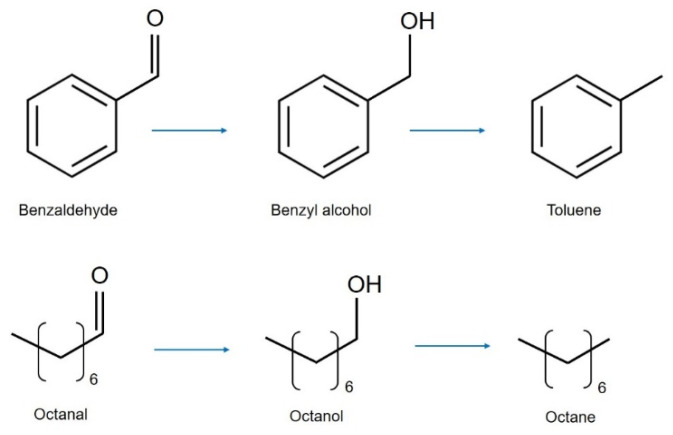
Reaction pathway of benzaldehyde (**top**) and octanal (**bottom**) hydrogenation/hydrogenolysis.

**Figure 4 nanomaterials-12-00908-f004:**
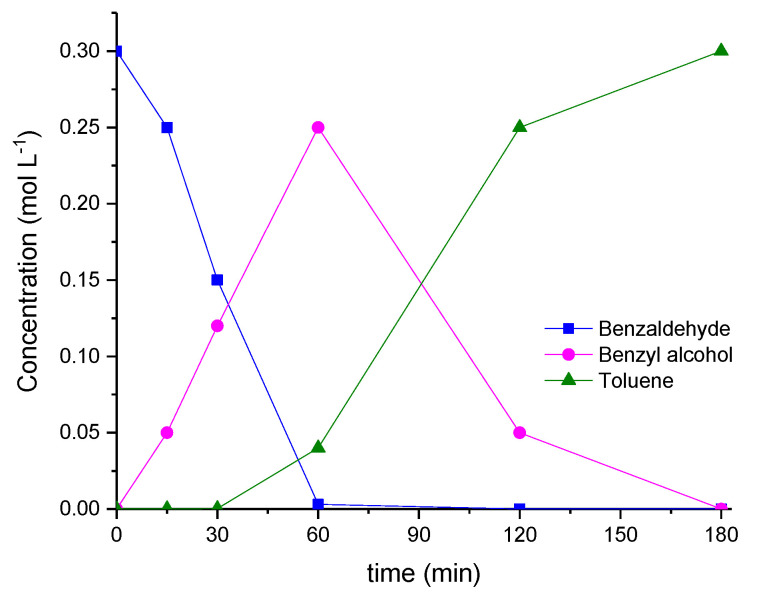
Benzaldehyde hydrogenation/hydrogenolysis reaction. Reaction conditions: 50 °C, 2 bar of H_2_, substrate-to-metal ratio 1000:1, benzaldehyde 0.3 mol L^−1^ in *p-*xylene.

**Figure 5 nanomaterials-12-00908-f005:**
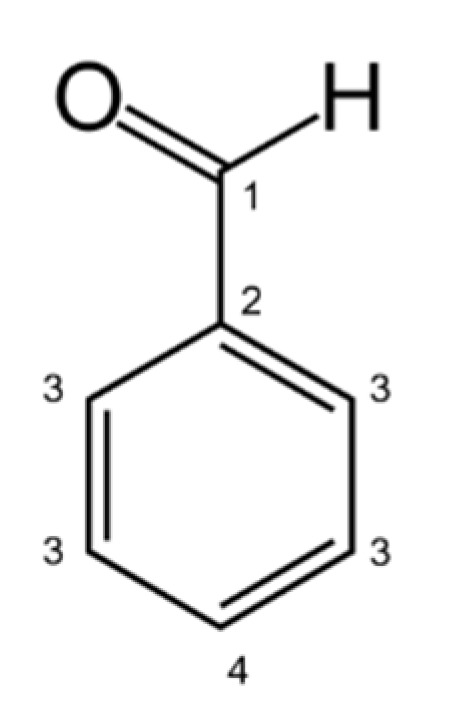
Differentiation of benzaldehyde carbon atoms detectable by T1 ^13^C NMR.

**Figure 6 nanomaterials-12-00908-f006:**
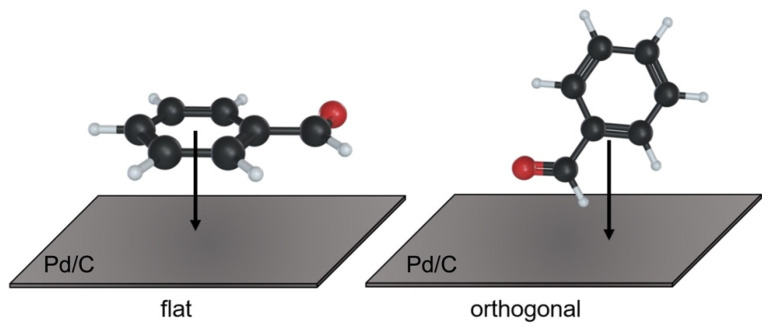
Different adsorption modes of benzaldehyde on Pd/C catalyst.

**Figure 7 nanomaterials-12-00908-f007:**
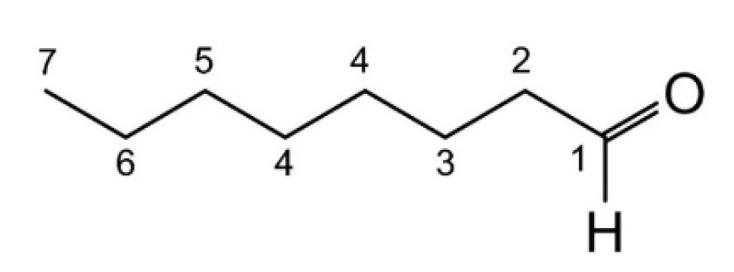
Differentiation of n-octanal carbon atoms detectable by T1 ^13^C NMR.

**Figure 8 nanomaterials-12-00908-f008:**
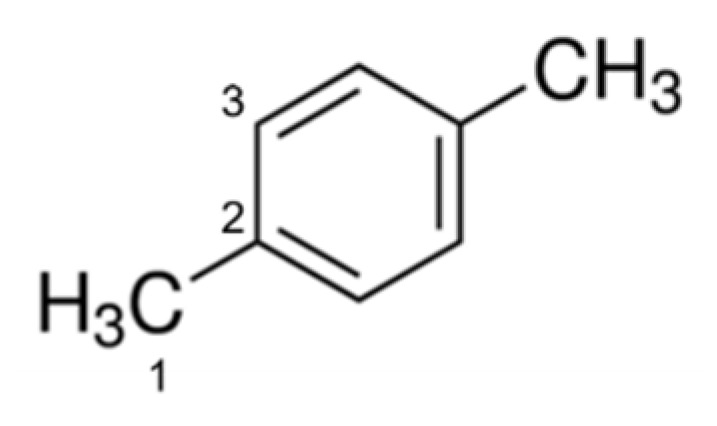
Differentiation of *p-*xylene carbon atoms detectable by T1 ^13^C NMR.

**Figure 9 nanomaterials-12-00908-f009:**
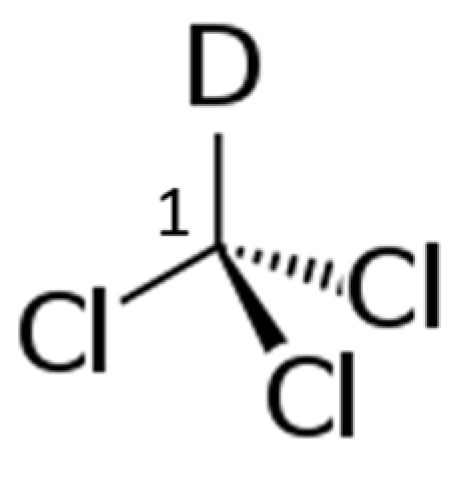
CDCl_3_ carbon atom detectable by T1 ^13^C NMR.

**Figure 10 nanomaterials-12-00908-f010:**
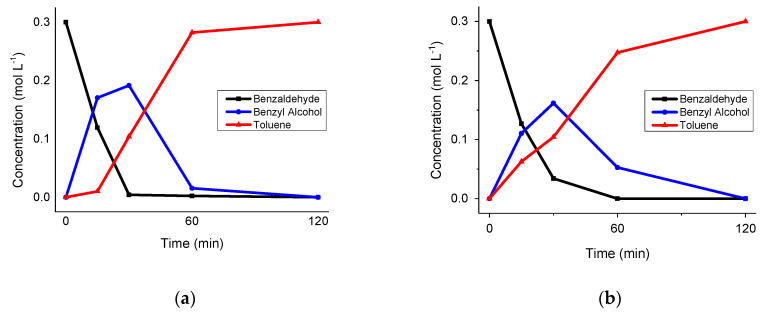
Benzaldehyde hydrogenation/hydrogenolysis reaction in *p-*xylene (**a**) or in dodecane (**b**). Reaction conditions: 50 °C, 2 bar of H_2_, substrate-to-metal ratio 1000:1, benzaldehyde 0.3 mol L^−1^ in *p-*xylene.

**Table 1 nanomaterials-12-00908-t001:** Characterization of the carbon support.

Carbon Support	BET Surface Area	Pore Volume	Oxygen Content (CHN)
GNP	490 m^2^ g^−1^	0.84 mL g^−1^	5.9%

**Table 2 nanomaterials-12-00908-t002:** XPS analysis of the Pd 3d orbital with Pd^0^ and Pd^2+^ relative abundance of 1% Pd/GNP.

	Pd 3d	
	Pd^0^ (%)	Pd^2+^ (%)
*B.E. (eV)*	335.8	337.0
At. %	68	32

**Table 3 nanomaterials-12-00908-t003:** Summary of the catalytic behaviour of 1% Pd/GNP towards aldehydes hydrogenation (50–150 °C and 2–20 bar of H_2_).

Substrate ^a^	Conversion at 5 h (50 °C, 2 bar H_2_)	Conversion at 5 h (150 °C, 20 bar H_2_)
Benzaldehyde	100%	-
Octanal	0%	0%

^a^ Reaction conditions: substrate-to-metal ratio 1000:1, substrate 0.3 mol L^−1^ in *p-*xylene.

**Table 4 nanomaterials-12-00908-t004:** Benzaldehyde in *p*-Xylene-d_10_: T1_bulk_ and T1_ads_ resulting from the analysis of the inversion recovery curves (Appendix A) from ^13^C NMR spectra in high-field (600 MHz). (C indicates each benzaldehyde carbon atom as in Figure 4).

Benzaldehyde		+GNP		+Pd-GNP	
	T1_bulk_	T1_ads_	T1_ads/_T1_bulk_	T1_ads_	T1_ads/_T1_bulk_
C^1^	15.1	13	0.86	9.88	0.65
C^2^	36.3	33	0.90	33.5	0.92
C^3^	10.5	10.4	0.99	7.82	0.74
C^4^	11.8	11.8	1.00	11.7	0.99
mean			**0.94**		**0.82**

**Table 5 nanomaterials-12-00908-t005:** Octanal in *p*-Xylene-d_10_: T1_bulk_ and T1_ads_ resulting from analysis of the inversion recovery curves (Appendix A) from ^13^C NMR spectra in high-field (600 MHz).

Octanal		+GNP		+Pd-GNP	
	T1_bulk_	T1_ads_	T1_ads/_T1_bulk_	T1_ads_	T1_ads/_T1_bulk_
C^1^	11.92	11.60	0.97	11.80	0.99
C^2^	6.24	5.95	0.95	5.87	0.94
C^3^	6.05	5.26	0.87	5.22	0.86
C^4^	4.83	4.84	1.00	4.73	0.98
C^5^	6.17	6.08	0.99	5.69	0.92
C^6^	5.48	5.01	0.91	5.24	0.96
C^7^	6.10	6.03	0.99	5.88	0.96
mean			**0.95**		**0.94**

**Table 6 nanomaterials-12-00908-t006:** *p*-Xylene T1_bulk_ and T1_ads_ resulting from analysis of the inversion recovery curves from ^13^C NMR spectra (600 MHz).

*p*-Xylene		+GNP		+Pd-GNP	
	T1_bulk_	T1_ads_	T1_ads/_T1_bulk_	T1_ads_	T1_ads/_T1_bulk_
C^1^	30.7	25.5	0.83	27.4	0.89
C^2^	36.8	27.5	0.75	28.2	0.77
C^3^	39.5	34.0	0.86	32.5	0.82
mean			**0.82**		**0.83**

**Table 7 nanomaterials-12-00908-t007:** Octanal and dodecane T1_bulk_ and T1_ads_ resulting from analysis of the inversion recovery curves from ^13^C NMR spectra (600 MHz).

		+GNP		+Pd-GNP	
	T1_bulk_	T1_ads_	T1_ads/_T1_bulk_	T1_ads_	T1_ads/_T1_bulk_
C^1^ (octanal)	12.0	12.0	1.0	10.3	0.85
C^1^ (dodecane)	4.02	3.95	0.98	3.95	0.98

**Table 8 nanomaterials-12-00908-t008:** Octanal in *CDCl_3_*: T1_bulk_ and T1_ads_ resulting from analysis of the inversion recovery curves (Appendix A) from ^13^C NMR spectra in high-field (600 MHz). * the value of 0.64 is very different from the ratio, possibly due to an overlap with impurities in the sample.

**(a) CDCl_3_**		**+GNP**		**+Pd-GNP**	
	**T1_bulk_**	**T1_ads_**	**T1_ads/_T1_bulk_**	**T1_ads_**	**T1_ads/_T1_bulk_**
C^1^	95.7	95.7	1	90.4	0.94
**(b) Octanal**		**+GNP**		**+Pd-GNP**	
	**T1_bulk_**	**T1_ads_**	**T1_ads/_T1_bulk_**	**T1_ads_**	**T1_ads/_T1_bulk_**
C^1^	11.7	11.0	0.94	10.8	0.87
C^2^	5.45	5.32	0.97	4.50	0.82
C^3^	5.28	5.20	0.98	4.50	0.85
C^4^	4.70	4.22	0.90	3.87	0.81
C^5^	6.19	6.00	0.97	3.97	0.64 *
C^6^	4.95	4.91	0.99	3.97	0.80
C^7^	5.92	5.90	0.99	n.d.	n.d.
mean			**0.96**		**0.80**

**Table 9 nanomaterials-12-00908-t009:** Results obtained in octanal hydrogenation in different solvents. Reaction occurred at 150 °C, 20 bar of H_2_ and Pd:octanal molar ratio of 1:500. Conversion has been calculated after 2 h.

Catalyst		Conversion %	Selectivity	
	Solvent		Octanol %	Dioctyl Ether %
Pd/GNP	*p-*xylene	3	100	0
Pd/GNP	dodecane	40	0	99

## Data Availability

Not applicable.

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
