# Peer review of "An Insight into the Role of Reactant Structure Effect in Pd/C Catalysed Aldehyde Hydrogenation"

_nanomaterials, 2022, doi:10.3390/nano12060908_

Round 1
Reviewer 1 Report
The work presented by the authors use the potential of 13C NMR T1 relaxation times as an indicator to describe the strength of the interaction between a reactant and the catalytic surface. I find the study of interest for many catalytic applications but the quality of the manuscript needs to be improved. More precisely, the authors include in their manuscript only the values of the T1 for the free and the adsorbed solvent but given the issues with the low abundance of 13C, it is very important to show the quality of the spectra they obtained and the T1 inversion-recovery curves to have a further proof of the 1% error stated for the extracted T1 values. I suggest thus to include these data in the manuscript and for selected solvents even the spectra aquired after different variable delays d2.
Minor things:
- page 2, introduction: the authors state that FTIR does not allow the study of the competition between the reactand and solvent for liquid ohase reactions. Could the authors possible add more information why this is not possible?
- Page 2, introduction: it is stated that ATR-IR is limited to aqueous environements. I was the meaning that ATR-IR can be very good conducted also in solid-state. More, usually, the IR is usually by far faster than NMR.
- The authors state that the T1 values are obtained from the integration of the 13C NMR spectra. The T1 values are obtained from the analysis of the inversion-recovery curves. These inversion-recovery curves are obtained from the integration of the 13C spectra.
- Define the meaning of d1 in the pulse sequence for inversion-recovery
- On page 5 top, please define the meaning of a in the equation.
Author Response
Reviewer 1
The work presented by the authors use the potential of 13C NMR T1 relaxation times as an indicator to describe the strength of the interaction between a reactant and the catalytic surface. I find the study of interest for many catalytic applications but the quality of the manuscript needs to be improved. More precisely, the authors include in their manuscript only the values of the T1 for the free and the adsorbed solvent but given the issues with the low abundance of 13C, it is very important to show the quality of the spectra they obtained and the T1 inversion-recovery curves to have a further proof of the 1% error stated for the extracted T1 values. I suggest thus to include these data in the manuscript and for selected solvents even the spectra acquired after different variable delays d2.
We thank the reviewer for her/his comment. We agree about the importance of showing the spectra and the T1 inversion-recovery curves, which have been added in the supporting information.
We thank the reviewer for his comment. We agree about the importance of showing the spectra and the T1 inversion-recovery curves, which have been added in the supporting information.
In particular, in SI we reported the pseudo-2D NMR spectra for T1 measurement (acquired with d2 range between 0.05 and 100s) and the analysis of the inversion recovery curve for the aldehyde carbon for Benzaldehyde in p-xylene (Fig. S1-S3) and for Octanal in p-xylene (Fig. S4-S6) and CDCl3 (Fig. S7-S9). The analyses of T1 inversion recovery curves allows the determination of the error in the range 0.6 -1 %.
Minor things:
- page 2, introduction: the authors state that FTIR does not allow the study of the competition between the reactant and solvent for liquid phase reactions. Could the authors possible add more information why this is not possible?
FTIR spectroscopy allows the characterization of the surface sites in catalysts using probe molecules. This allows studying the different active site characteristic but not their interaction with the molecules involved into the reaction. With NMR analyses we look at the interaction of the real molecule we are interested in with the catalytic surface.
- Page 2, introduction: it is stated that ATR-IR is limited to aqueous environments. I was the meaning that ATR-IR can be very good conducted also in solid-state. More, usually, the IR is usually by far faster than NMR.
ATR-IR is typical for studying the solid-gas or solid-liquid interfaces. However, it is not applicable to carbon materials. ATR-IR is indeed not limited at aqueous environment (there was a mistake in the paper). We modified the text accordingly.
- The authors state that the T1 values are obtained from the integration of the 13C NMR spectra. The T1 values are obtained from the analysis of the inversion-recovery curves. These inversion-recovery curves are obtained from the integration of the 13C spectra.
We thank the reviewer for this correction. Accordingly, we changed the captions of each table as
“..T1bulk and T1ads resulting from analysis of the inversion recovery curves from 13C NMR spectra in high-field (600 MHz).”
- Define the meaning of d1 in the pulse sequence for inversion-recovery
d1 is the relaxation delay. At pag. 5, lines 205-207 we improved the sentence
“14 T1 recovery delays were used ranging from 0.05 ms to 100 s. We used 8 repeat scans and a relaxation delay (d1) of 50 s between each scan to ensure a maximum signal (and to ensure the reach of the equilibrium, the d1 delay in the pulse sequence should be set to ~5 * the longest T1 of interest in the molecule) was maintained at all times.”
- On page 5 top, please define the meaning of a in the equation.
We explained the meaning of a in the main text.
Reviewer 2 Report
This paper was a joy to read. A lovely study, well executed and well reported. I have no hesitation in recommending acceptance.
Author Response
This paper was a joy to read. A lovely study well executed and well reported. I have no hesitation in recommending acceptance.
We sincerely thank the reviewer for having appreciate our work and for the kind comment.
Reviewer 3 Report
In this article, Stucchi, Prati and coworkers focused on the catalyst of Pd nanoparticles supported on GNP, which were used for the hydrogenation of benzaldehyde and octanal aldehydes. The different activity of a 1%Pd/carbon catalyst towards aromatic and aliphatic aldehydes hydrogenation has been explored by 13C NMR relaxation. They used the ratio between T1 relaxation times of adsorbed (ads) and free diffusing (bulk) molecules (T1ads/T1bulk) to indicate the relative strength of interaction between the reactant and the catalytic surface. They proved that by using 13C T1 NMR, this difference in catalytic behaviour can be predominantly attributed to the contrasting adsorption behaviour of both substrates on the catalytic surface which is strongly affected by the choice of solvent. The NMR results were able to guide the selection of more suitable reaction conditions.
In my opinion, this study is appropriate for the publication in Nanomaterials after the catalyst (1%Pd/GNP) has been well characterized, such as SEM?
Author Response
In my opinion, this study is appropriate for the publication in Nanomaterials after the catalyst (1%Pd/GNP) has been well characterized, such as SEM?
We thank the reviewer for the suggestion. Pd/GNP was characterized by TEM and XPS, and we now added these characterizations in the manuscript, as follows:
“The catalyst was characterized by TEM showing a metal particle dimension and distribution of 3.5 ± 1.1 nm (Fig. 2) and then by XPS (Tab. 2) showing a Pd0 and Pd2+ relative abundance of 68 % and 32 % respectively.
The catalytic activity was evaluated in benzaldehyde and octanal hydrogenation, for comparison between an aromatic and an aliphatic aldehyde, using the same catalyst thus allowing considering only the effect of the different substrate.”
All the additions or modifications in the main text are highlighted in yellow.
Reviewer 4 Report
An insight into the role of reactant structure effect in Pd/C
catalyzed aldehyde hydrogenation
nanomaterials-1613244
In this manuscript the authors describe the use of the measure of relaxation times of organic molecules by 13C NMR in the presence or not of a based metal catalyst or the corresponding support. If the relaxation times of the molecules are higher in the presence of the catalysts (or support) indicate an adsorption with it and its strength. This adsorption is governed by the nature of the catalytic substrate and/or the solvent.
The article is well-written, the bibliography appropriate. I have some concerns about the interest of the readers about the paper, as the molecules chosen are not of the highest interest and the methodology has been described before. I do not understand why the authors decide to present the results in three parts. In my opinion, the “proof of concept” section should not exist, and all catalytic results should be presented in the same section, and after the NMR section, which allow to explain the catalytic results. It could be appropriate to add a paragraph in order to know when a T1ads/T1bulk is significant enough to talk about interaction, and its strength.
I think if the authors change the distribution of the results and add some precisions about the T1 ads/ T1 bulk it could be published at nanomaterials.
Some minor comments
Usually the condition in which catalysis have been performed are given in the tables presenting the catalytic results.
It could be interesting to add some nmr spectra into the paper.
In table 7 a T1ads/T1bulk of 0.64 is given for octanal C5. Is there a mistake in the data? If not thare is an explanation?
Can the authors add some theoretical calculations to confirm the experimental data about adsorption?
Author Response
The article is well-written, the bibliography appropriate. I have some concerns about the interest of the readers about the paper, as the molecules chosen are not of the highest interest and the methodology has been described before.
We thank the reviewer for his worthwhile comments and suggestions.
Concerning the chosen molecules, they are crucial for having model indicators to study the catalytic hydrogenation/hydrogenolysis of aldehydes. These reactions are really of high interest from an industrial point of view and particularly relevant in the field of biomass transformation.
Benzaldehyde and n-octanal are examples of aromatic and of aliphatic carbonyl group and they can be used to establish the influence of the aldehyde structure on the behavior of the catalyst. It is well known that the carbonyl group of aliphatic and aromatic aldehyde differs from an electronic point of view but this study established that also the molecular structures play a fundamental role in determining the catalytic activity. Moreover, it has been established that the competition between reactant and solvent is fundamental.
The following sentence has been added to the Introduction:
“The catalytic hydrogenation/hydrogenolysis of aldehydes is of high interest from an industrial point of view and particularly relevant in the field of biomass transformation. Indeed, the high oxygen content present in the biomass must be removed to develop biomass-based processes and hydrogenation/hydrogenolysis catalytic processes represent one of the most appealing alternatives. Benzaldehyde and n- octanal can be considered model molecules that can be used to establish the influence of the aldehyde structure on the behavior of the catalyst.”
NMR relaxation is already reported, but it has been now stated that the use of only T1 (instead of T1 and T2) with the advantage of excluding the effect of coupling constants. Moreover, the use of 13C -NMR allows to follow the signal of each C of the molecules whereas using 1H-NMR could be more difficult.
I do not understand why the authors decide to present the results in three parts. In my opinion, the “proof of concept” section should not exist, and all catalytic results should be presented in the same section, and after the NMR section, which allow to explain the catalytic results.
We thank the Reviewer for her/his suggestion. Following the advice, we merged the “proof of concept” paragraph with the previous section.
It could be appropriate to add a paragraph in order to know when a T1ads/T1bulk is significant enough to talk about interaction, and its strength.
As required by the reviewer, we added a paragraph in the main text, in the paragraph 3.2, as follows:
“In particular, a T1ads/T1bulk ratio equal or very close to 1 indicates that there is no interaction and thus no adsorption of the substrate on the catalysts, as well as T1ads/T1bulk values ≥ 0.9 indicate a very weak interaction and negligible adsorption.”
I think if the authors change the distribution of the results and add some precisions about the T1 ads/ T1 bulk it could be published at nanomaterials.
As already mentioned, we merge the additional paragraph entitled “proof of concept”, including it in the NMR section because this further reaction has been used to verify the results obtained by NMR.
Some minor comments
Usually the condition in which catalysis have been performed are given in the tables presenting the catalytic results.
We reported the reaction conditions as legend of Table as suggested by the Reviewer.
It could be interesting to add some nmr spectra into the paper.
We added the pseudo-2D NMR spectra in SI for not affecting the readability of the paper. In particular we reported the pseudo-2D NMR spectra for T1 measurement and the analysis of the inversion recovery curve for the aldehyde carbon for Benzaldehyde in p-xylene (Fig. S1-S3) and for Octanal in p-xylene (Fig. S4-S6) and CDCl3 (Fig. S7-S9).
In table 7 a T1ads/T1bulk of 0.64 is given for octanal C5. Is there a mistake in the data? If not there is an explanation?
The value is correct even if it is slightly off the scale compared to all the others. This can be due to the presence of small amount of octanoic acid (derived from the oxidation of the octanal in the NMR tube) and to the overlapping of the resonance of C5. We explained this point in a note in the table caption.
Moreover, we calculate the ratio for all carbons since the most reliable interpretation of such T1ads/T1bulk values is to use the mean T1ads/T1bulk ratio as an indicator of the interaction between reactant and catalyst surface.
As the mean T1ads/T1bulk was 0.80, octanal is adsorbed on Pd/GNP when CDCl3 was used as the solvent, and consistently, the T1ads/T1bulk value of each carbon atom indicates that there is adsorption.
Can the authors add some theoretical calculations to confirm the experimental data about adsorption?
Some DFT calculation of octanal adsorption on carbon-based catalysts showed that, normalizing the adsorption energies to the number of molecules per surface area of Pd, octanal binds more strongly to the Pd surface than benzaldehyde [see ref. 20, S. Cattaneo et. al, Journal of Catalysis 399 (2021) 162–169]. However, the differences in adsorption energy revealed between the two substrates can’t justify the complete inactivity of octanal hydrogenation on the Pd/C catalyst. The reason of this discrepancy comes from the role of the solvent, not taken into account in the theoretical calculation but strongly highlighted by NMR. Theoretical calculation would become excessively complex by including solvent molecule which however can be crucial to disclose the reason of a certain catalytic behavior. Therefore, in our opinion theoretical calculations present the limitation of not considering the whole catalytic system.
Reviewer 5 Report
The manuscript by L. Prati et al. describes the use of 13C NMR relaxation to investigate interactions of substrates and solvents with the surface of solid catalysts, and their relation to catalytic performance observed experimentally. The authors selected the hydrogenation of benzaldehyde and octanal using a Pd/carbon catalyst as model reaction, and could evidence a clear link between the adsorption strengths of substrates and solvents on the catalyst surface and the resulting catalytic activity.
The preparation, characterization and catalytic results presented in this study are convincing, well described and important, making this work a valuable contribution to the field.
Thus, I recommend acceptance of this manuscript for publication in Nanomaterials.
However, I would strongly suggest to provide the NMR spectra in a Supplementary Information file (or at least, a few representative NMR spectra).
Author Response
However, I would strongly suggest to provide the NMR spectra in a Supplementary Information file (or at least, a few representative NMR spectra).
We sincerely thank the reviewer for having appreciated our work and for her/his comments. As required, in SI we reported the pseudo-2D NMR spectra for T1 measurement and the analysis of the inversion recovery curve for the aldehyde carbon for Benzaldehyde in p-xylene (Fig. S1-S3) and for Octanal in p-xylene (Fig. S4-S6) and CDCl3 (Fig. S7-S9).
Round 2
Reviewer 1 Report
The authors commented on all my requirements and added the required exprimental data for the T1. I support the publication of this version of the manuscript.